# Development and Application of a Visual Duck Meat Detection Strategy for Molecular Diagnosis of Duck-Derived Components

**DOI:** 10.3390/foods11131895

**Published:** 2022-06-26

**Authors:** Xiaoyun Chen, Huiru Yu, Yi Ji, Wei Wei, Cheng Peng, Xiaofu Wang, Xiaoli Xu, Meihao Sun, Junfeng Xu

**Affiliations:** 1State Key Laboratory for Managing Biotic and Chemical Threats to the Quality and Safety of Agro-Products, Zhejiang Academy of Agricultural Sciences, Hangzhou 310021, China; xiaoyunchen_2016@163.com (X.C.); yhr_yee@163.com (H.Y.); jymemory12138@163.com (Y.J.); weiw8426@163.com (W.W.); pc_phm@163.com (C.P.); yywxf1981@163.com (X.W.); xuxiaoli@zju.edu.cn (X.X.); 2College of Chemistry and Life Sciences, Zhejiang Normal University, Jinhua 321001, China

**Keywords:** RPA, PCR, colloidal gold immunoassay strips

## Abstract

To make meat adulteration detection systems faster, simpler and more efficient, we established a duck-derived meat rapid detection Recombinase Polymerase Amplification (dRPA) method by using interleukin 2 (IL-2) from nuclear genomic DNA as the target gene to design specific primers. We tested the dRPA detection system by comparing its sensitivity and specificity using real-time fluorescent PCR technology. By adjusting the ratio of reagents, this method shortens the time of DNA extraction and visualizes results in combination with colloidal gold immunoassay strips. Our results demonstrate that this dRPA method could specifically detect duck-derived components with a sensitivity of up to 23 copies/μL and the accuracy of the results is consistent with real-time fluorescent PCR. Additionally, dRPA can detect at least 1% of the duck meat content by mixing beef and mutton with duck meat in different proportions, which was verified by spot-check market samples. These results can be visualized with colloidal gold immunoassay strips with the same accuracy as real-time fluorescent RPA. dRPA can complete detection within 30 min, which shortens existing detection time and quickly visualizes the detection results on-site. This lays the groundwork for future large-scale standardized duck origin detection.

## 1. Introduction

Meat is an important food with a protein content of 20–35% that provides the essential amino acids for humans [1] and also contains necessary minerals, fats, fatty acids and vitamins, all of which contribute to human health [2]. However, as living standards improve and meat demand increases, some merchants mix low-quality meat with high-priced meat to reap large profits. Some scandals have been widely reported in recent years, including fake beef and mutton made with duck meat and fake beef jerky made with 100% pure duck meat. It is common to pass off duck meat as beef and mutton, especially in barbecue stalls, food stalls and hot pot restaurants. The first step in “transforming” duck meat into beef and mutton is to apply butter or suet oil on the duck skewers and use seasonings to mask the original taste to confuse the consumer. Another method is to soak chopped duck meat in additives bought from a market, such as “beef paste” and “mutton essence”. These make it difficult for consumers to judge the meat simply by the taste, threatening consumer health and seriously disrupting the market order. Therefore, it is necessary to establish an efficient, fast and accurate method for detecting duck meat.

To date, several identification methods have been developed, most of which detect species-specific proteins or DNA [3]. Protein-based methods, such as enzyme-linked immunosorbent assay and lateral flow immunoassay [4,5,6,7,8,9], rely on detecting peptide targets, which are always tissue-dependent [10]. Unlike protein-based methods, DNA-based methods have a high degree of stability. Therefore, they can analyze fresh, processed and adulterated meat products by detecting small amounts of DNA [11,12]. Therefore, these DNA-based methods, including PCR, multiplex PCR, digital PCR and RPA, have been widely used in species identification for meat adulteration detection [13,14]. However, due to various limitations, such as the high requirements of PCR instruments for operators, the strict requirements of LAMP technology for primer design and the difficulty of transporting experimental equipment, it is important to identify a faster visual detection method for detecting meat adulteration.

This study aims to establish a method for rapid visual detection of meat adulteration based on RPA. RPA is a new in vitro nucleic acid isothermal amplification technology that can amplify the desired target fragments under constant temperature (37–42 °C) for 20 min [15]. This technology does not require large-scale experimental instruments, is simple to operate and provides specific results, all of which make it a new type of nucleic acid amplification technology that can replace traditional PCR technology [16,17]. So far, RPA technology has been used in many detection fields [18,19,20,21,22]. In contrast, traditional RPA technology still relies on agarose gel electrophoresis to observe the amplification results, which is a complicated and time-consuming process. Therefore, in this study, duck, mutton and beef were used to simulate meat adulteration and real-time fluorescent RPA was used for detection. This not only improved accuracy but served as a control for the subsequent visualization research results and increased the reliability of these results. To realize the on-site visualization of the results, we combined real-time fluorescent RPA technology with colloidal gold immunoassay strips, used double-antibody sandwich technology to prepare the test strips and combined the diluted amplification products with the test strips. After these measures, the experimental results can be directly observed with the naked eye.

## 2. Materials and Methods

### 2.1. Meat

The fresh meat used in the experiment (duck, beef, mutton, chicken, pork, fox, mouse, horse, venison, dog, pigeon, quail and goose) came from the Zhejiang Academy of Agricultural Sciences Institute of Animal Husbandry and Veterinary Medicine and Beijing Institute of Animal Husbandry and Veterinary Medicine, Chinese Academy of Agricultural Sciences. The processed meat used for market testing experiments (braised duck liver, braised duck heart, braised duck neck, braised duck gizzard, raw duck neck, raw duck heart, raw duck gizzard, braised beef, fat sheep roll, fat beef rolls, beef balls) was purchased from local markets.

### 2.2. DNA Extraction and Sample Preparation

Conventional DNA extraction typically takes a long time. To shorten the experimental time, we achieved one-step DNA extraction by adjusting the content of each component in the lysate (Table 1). We added about 25 mg of chopped animal tissue into a 1.5 mL centrifuge tube, then added 150 μL of lysate, shook and mixed it and placed it in a water bath at 95 °C for 5 min. The reacted lysate was amplified for direct fluorescence at a constant temperature and the remaining sample nucleic acid was stored in a refrigerator at −20 °C for future use.

To simulate market adulteration, we mixed the DNA extracted from beef, mutton and duck meat at different ratios (Table 2), which were used as samples for subsequent experiments.

### 2.3. Primer and Probe Design

In this experiment, IL-2 in nuclear genomic DNA was selected as the target gene to design primers for the specific detection of duck meat by RPA reaction. IL-2 is a cytokine with an immunoregulatory effect secreted by lymphocytes after being stimulated by antigen, plays an important role in the body’s immune response and antiviral infection and is a single-copy gene. IL-2 of different species has its species-specific sequence. Since the sequence alignment of the IL-2 gene from different duck breeds was identified, this specific sequence exists in all the collected ducks from different subfamilies. Figure 1 displays the gene sequence. The probe sequence has a total of 37 bases. Its 22nd position is FAM-dT, the 25th position is the abasic site of THF and the 27th position is BHQ-dT, 3′ phosphorylated. All primers and probes were synthesized by Invitrogen Biotechnology Co., Ltd. (Shanghai, China) (Table 3).

### 2.4. Reaction System

#### 2.4.1. PCR

A 25 μL volume PCR reaction mixture consisting of 12.5 μL 2 × TaqMan Universal Master mix (Applied Biosystems, Foster City, CA, USA), 8 μL ddH_2_O, 2 μL DNA template, 1 μL forward and reverse primers and 0.5 μL probe was performed. PCR reactions were performed on the Bio-Rad CFX96 PCR Detection System (Bio-Rad Company, Pleasanton, CA, USA). The PCR program consisted of an initial denaturation step (95 °C/5 min), followed by 1.5 h at 95 °C/10 s and 58 °C/32 s to amplify the desired gene.

#### 2.4.2. RPA

The RPA reaction was performed according to the manufacturer’s instructions (DNA Isothermal Rapid Amplification Kit, Fluorescent Type, Huaifang Amp Future, ChangZhou, China). The total reaction volume was 50 μL, which contained 29.4 μL A buffer, 2 μL DNA template, 2.5 μL B buffer, 2 μL forward and reverse primers, 0.6 μL probe and 11.5 μL ddH_2_O. This reaction was performed on a Bio-Rad CFX96 PCR reaction detection system and the results were analyzed using CFX Manager 3.1 software (Bio-Rad, Berkeley, CA, USA). The RPA reaction was performed at temperatures ranging from 37 °C to 42 °C for 20 min (the fluorescence intensity was read and recorded every 30 s).

#### 2.4.3. Screening of the Best Primer Pair for RPA Reaction

To obtain the best combination of primers with probes, all reverse primers (R1–R5) and one forward primer (F1) with probes were first used in the experiment to select the best reverse primer and then the primer was used to screen all forward primers. The optimal primer pair was evaluated and determined based on the time at which amplification started and the fluorescence signal intensity of each RPA reaction for various primer and probe combinations.

### 2.5. Optimization of RPA

The settings of primer-probe concentration, experimental temperature and time are extremely important for the reaction. Different primer-probe concentrations, too-high or too-low reaction temperature and too-long or too-short reaction time will affect the stability and accuracy of the experimental results. The meat DNA template was diluted to a certain concentration by adjusting the concentration ratio of primers and probes, using the RPA amplification instructions to identify the optimal reaction temperature (37 °C to 42 °C) and choosing different primer pairs. Amplification was performed under different primer-probe concentrations, temperatures and times. The intensity of the fluorescent signal was analyzed to identify the optimal primer-probe concentration, reaction temperature and reaction time after the reaction.

### 2.6. Specificity and Sensitivity of dRPA Reaction

The specificity of the dRPA assay was assessed by testing all animal tissue materials. Serial dilutions of duck DNA were performed to determine the limit of detection (LOD) of the method and the dilutions of genomic DNA contained approximately 5780 copies/μL, 1926 copies/μL, 642 copies/μL, 214 copies/μL, 71 copies/μL and 23 copies/μL.

### 2.7. Rapid Detection of RPA Reaction

Some instruments used in traditional testing are not common in daily life and remain in laboratory use, which restricts rapid on-site detection. Therefore, in this study, we used a quick extraction reagent combined with a colloidal gold immunoassay strip to study the rapid detection on site. To shorten the DNA extraction time, a quick-extraction reagent was used. Its formula reagent can extract DNA in only one step within 5 min at a 95 °C water bath by adjusting the concentration of EDTA, Tris-HCl, SDS, NaCl and proteinase K. This shortens the time required for the experiment and is convenient for the RPA reaction used in on-site detection. The reaction occurred at 39 °C for 20 min and finally, 5–10 μL amplified product was diluted 20 times with ddH_2_O in a centrifuge tube. After mixing, 50 μL was dropped into the sample well and the detection results of the reading area were recorded within 5 min. The whole reaction only takes only 30 min to produce results.

## 3. Results and Discussion

### 3.1. Primer and Probe Selection for Fluorescent RPA

Effective RPA results depend on the primers. Therefore, to obtain the best detection results, the optimal primer pair and probe combination for the recombinase polymerase chain reaction was selected by testing various primer combinations (Figure 2A,B). Forward primer F1 was tested with each reverse primer (R1–R5) in the RPA reaction. All reaction combinations had fluorescent signals (Figure 2A) and the F1/R4 combination had the best amplification results, with final fluorescence intensity of approximately 1450. Similarly, by analyzing the amplification effect of the reverse primer R4 combined with F1, F2, F3, F4 and F5, respectively (Figure 2B), the R4/F4 combination obtained the best amplification result, with final fluorescence intensity of approximately 1350. Based on these results, R4/F4 was selected as the best primer combination in the RPA reaction.

### 3.2. Optimization of Fluorescent RPA Reaction Conditions

#### 3.2.1. Optimization of Primer-Probe Concentration

Using duck meat as the detection sample, different primer-probe concentrations were set (Table 4) to optimize the final primer-probe concentration in the fluorescent RPA reaction system. After analyzing the fluorescence value of the reaction, 0.4 μM/L primer and 0.12 μM/L probe were considered the best concentration for the subsequent tests (Figure 3) and RPA amplification was performed on a CFX96 fluorescence PCR instrument (Bio-rad, Hercules, CA, USA) with the confirmed reaction system (Table 5).

#### 3.2.2. Fluorescence RPA Reaction Temperature Optimization

Six different temperatures were designed for screening to obtain the optimal reaction temperature. Results showed that the RPA amplification reaction had the best amplification curve and the highest fluorescence intensity at 39 °C (Figure 4). Therefore, 39 °C was the optimal reaction temperature for the subsequent RPA.

#### 3.2.3. Optimization of Fluorescence RPA Reaction Time

Reaction time affects the results and efficiency of the amplification. To confirm the optimal reaction time, two reaction times were set within the temperature range required for the reaction. The results (Figure 5A,B) demonstrated that RPA had a secondary amplification at about 30 min (Figure 5A), which failed to meet the experimental expectations. However, a better amplification curve was observed when the reaction time was approximately 20 min (Figure 5B). Therefore, 20 min was selected as the fluorescent RPA reaction time in this study.

### 3.3. Sensitivity and Specificity of Fluorescent RPA Detection

Species-specific experiments were conducted using duck, beef, mutton, chicken, pork, fox, mouse, horse, venison, dog, pigeon, quail and goose, with three replicates for each sample. The results demonstrated that the fluorescence intensities of duck meat samples all exceeded 300, while that of other meat samples were all below 50. This indicates that RPA could effectively distinguish duck meat from other meats with sufficient specificity (Figure 6).

The duck meat sample DNA used in the experiment is a duck-derived reference material independently developed by our laboratory, which has obtained the national secondary reference material certificate. Its copy number is 5780 copies/μL. We evaluated the sensitivity of this method by testing several different dilutions of duck DNA. As shown in Figure 7, when the dilution factor is less than or equal to 243 times, the fluorescence signal can be seen; when the dilution factor reaches 729, there is no obvious fluorescence signal. Therefore, the minimum LOD is 23 copies/μL.

### 3.4. Meat Blending Ratio Detection

Eight groups of experimental samples were prepared by mixing the DNA of beef and mutton, with duck DNA at ratios of 1%, 5%, 10% and 50%, to verify that the RPA reaction can be used to detect adulterated meat in practical settings. Fluorescent signals were detectable in all sample groups. The amplification curves of these mock samples are shown in Figure 8A,B. At the same time, we performed PCR reactions on the spiked samples (Figure 8C,D) to verify the RPA results. The high similarity of the two methods demonstrated the accuracy of the RPA reaction in detecting duck meat. In addition, we evaluated sensitivity by testing mixed samples with different adulteration ratios. The fluorescence signal can also be detected in 1% mixed samples and the fluorescence units (RFU) of beef-duck and mutton-duck are approximately 100 and 300, respectively. When the adulteration ratio is less than 5%, it is difficult to generate any commercial profits. Therefore, the minimum adulteration ratio in this experiment is 1%, which confirms the accuracy of the experimental method [23,24].

### 3.5. Market Actual Sample Testing

A variety of beef, mutton and duck products purchased from local markets (braised duck liver, braised duck heart, braised duck neck, braised duck gizzard, raw duck neck, raw duck heart, raw duck gizzard, braised beef, beef roll, fat sheep rolls, beef balls) were tested for practical application of the RPA reaction (Figure 9A) and a qPCR reaction was used as a validation (Figure 9B). As seen from the results, this technology can accurately distinguish duck-derived meat from other animal-derived meats in both frozen raw products and braised cooked food.

### 3.6. Rapid Detection of Duck-Derived Components with RPA Test Strips

#### 3.6.1. Test Strip Specificity Detection

To practically perform rapid and accurate on-site detection, we used a one-step rapid DNA extraction method, combined with colloidal gold test strips, to detect samples (Figure 10). Both the duck meat standard material and the Shaoxing duck meat DNA test results had double bands and were identified as positive. The beef DNA, mutton DNA and water all had a single band and were identified as negative. This result effectively confirmed that the specificity of the test strip in the field rapid detection is consistent with the research results of laboratory RPA and PCR reactions and demonstrated that the one-step DNA rapid extraction method combined with the test strip experiment was used in field detection.

#### 3.6.2. Test Strip Sensitivity Detection

The duck DNA samples were serially diluted and eight samples were combined with duck meat standard material and water. The amplified products after the RPA reaction were diluted and added to colloidal gold test strips for color development. The results of the combined test strip experiment after the RPA reaction (Figure 11) demonstrated that duck meat standard substance, duck meat DNA, 3-fold dilution, 9-fold dilution, 27-fold dilution, 81-fold dilution and 243-fold dilution samples all showed double bands, while the color of the band gradually weakened as the dilution ratio increased. This indicates that the test strip experiment is sensitive and can effectively identify and analyze whether the sample is positive. This method visualizes the results, making them clearer, easier to understand and simpler to operate, all of which contribute to rapid field detection.

#### 3.6.3. Test Strip Detection of Adulterated Sample

Market adulteration was simulated by mixing duck DNA into mutton and beef at different ratios. As shown in Figure 12A,B, all samples were positive; the lower the duck: meat ratio, the lighter the color of the test line. The experimental comparison demonstrates that positives can still be detected in mixed samples adulterated with 1% duck meat. This result is consistent with the above RPA reaction results, suggesting that the test strip test is both accurate and sensitive.

## 4. Conclusions

The results of the primer screening, specificity and sensitivity tests and PCR methods demonstrated that primers designed according to IL-2 are specific to duck-derived components. Only target fragments can be amplified with duck-derived components as templates; those with ox, sheep, chicken, pig, fox, mouse, horse, deer, dog, pigeon, quail and goose-derived templates cannot. After diluting the sample, the lowest LOD of duck-derived components was approximately 23 copies. When simulating meat adulteration, the duck-derived ingredients could still be detected at a 1% duck: meat ratio and the corresponding fluorescence results could be obtained by testing 11 market samples, which indicates that it can accurately detect the presence of adulterated meat. The comparison of dRPA and other methods of detection was show in Table 6. We also used a one-step DNA rapid extraction method and colloidal gold immunoassay strips to visualize RPA during the rapid, on-site detection of adulterated meat, simplifying the process. When appropriate concentration ratios of EDTA, Tris-HCl, SDS, NaCl and proteinase K are used, meat DNA can be extracted within 5 min and there is no need for large-scale experimental equipment. This means that detection can be performed quickly and easily. Additionally, the colloidal gold immunoassay strips can effectively visualize the results. Compared with the typical gel-running steps following a RPA reaction, this method shortens the time needed to generate results, is easier to operate and provides results that are easy to understand, making the on-site detection of meat adulteration simple and easy.

## Figures and Tables

**Figure 1 foods-11-01895-f001:**
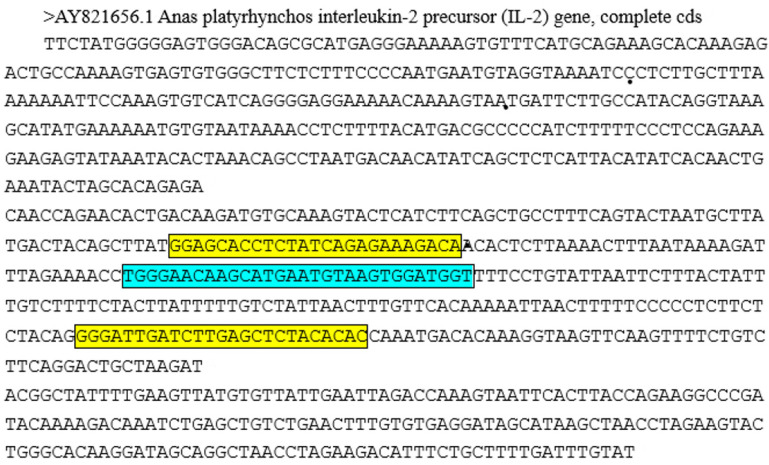
IL-2 gene sequence and primer-probe design location (Note: the yellow box area is the primer sequence and the blue box area is the probe sequence).

**Figure 2 foods-11-01895-f002:**
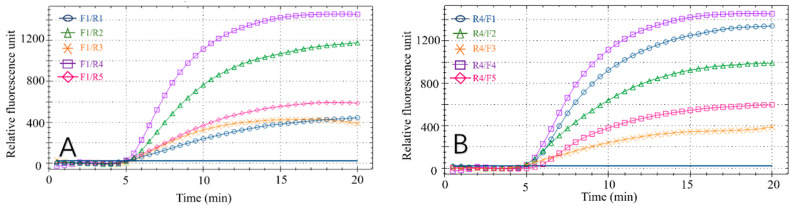
Results of primer screening. (**A**) Amplification results of forwarding primer F1 and all reverse primers. (**B**) Amplification results of reverse primer R4 and all forward primers.

**Figure 3 foods-11-01895-f003:**
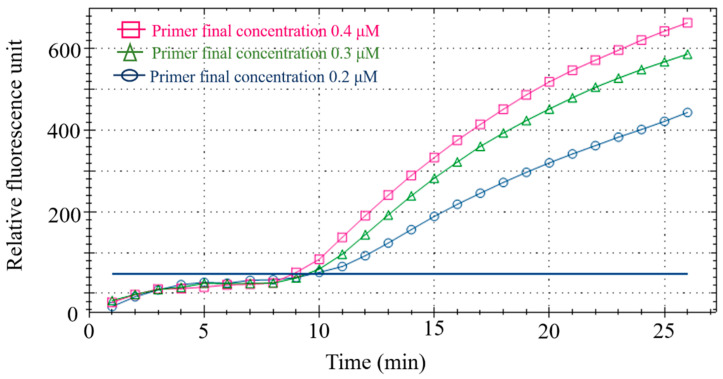
Primer-probe concentration optimization.

**Figure 4 foods-11-01895-f004:**
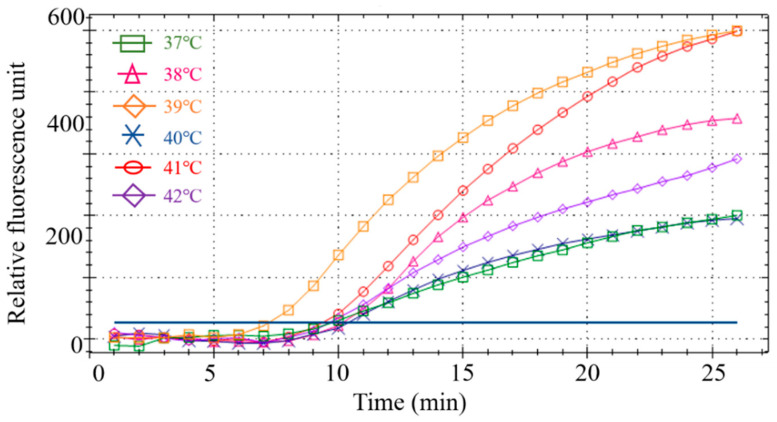
Screening of the optimal temperature for the RPA reaction.

**Figure 5 foods-11-01895-f005:**
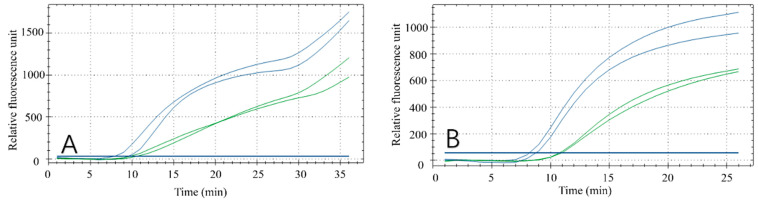
Optimization of reaction time. The green curve was performed with low concentrations of duck meat and the blue curve was performed with high concentrations of duck meat at 30 mins (**A**) and 35 mins (**B**).

**Figure 6 foods-11-01895-f006:**
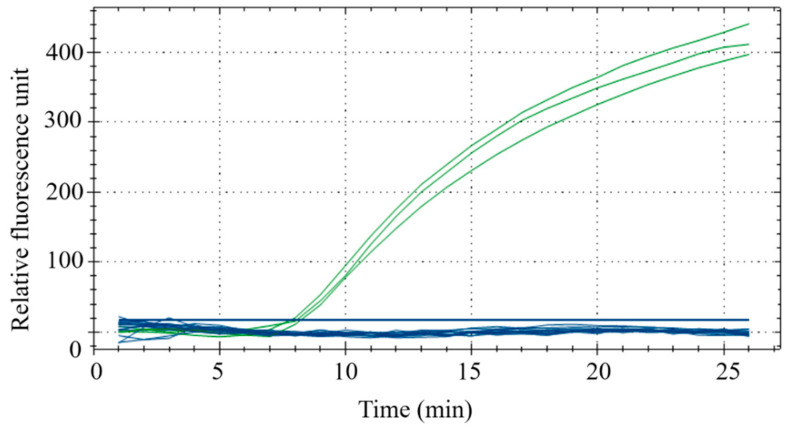
Interspecies specificity experiments. The samples for the green curve are duck meat and samples for the blue samples are beef, mutton, chicken, pork, fox meat, mouse meat, horse meat, venison, dog meat, pigeon meat, quail meat and goose meat.

**Figure 7 foods-11-01895-f007:**
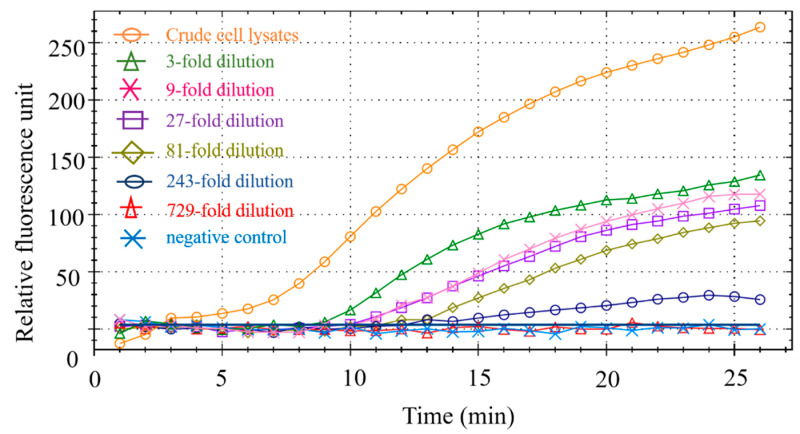
RPA sensitivity.

**Figure 8 foods-11-01895-f008:**
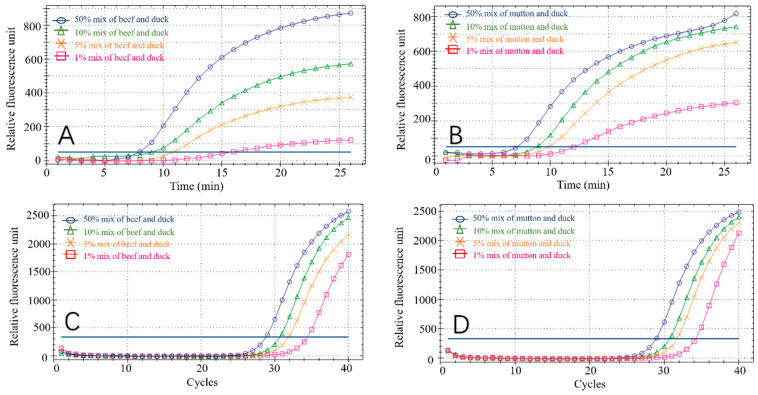
RPA detection results in mixed adulteration ratio for beef–duck (**A**), mutton–duck (**B**), beef–duck (**C**) and mutton–duck (**D**).

**Figure 9 foods-11-01895-f009:**
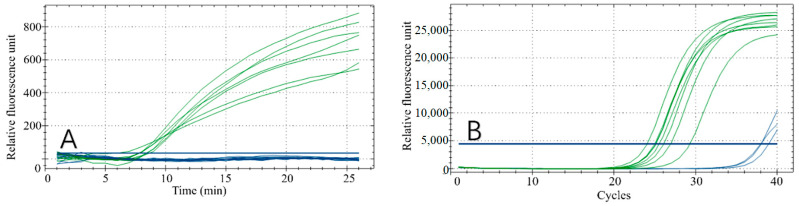
Market sampling results. RPA reaction results (**A**); qPCR reaction results (**B**). Samples for the green curves are braised duck liver, braised duck heart, braised duck neck, braised duck gizzard, raw duck neck, raw duck heart and raw duck gizzard. The samples for the blue curves are braised beef, beef roll and sheep rolls, beef balls.

**Figure 10 foods-11-01895-f010:**
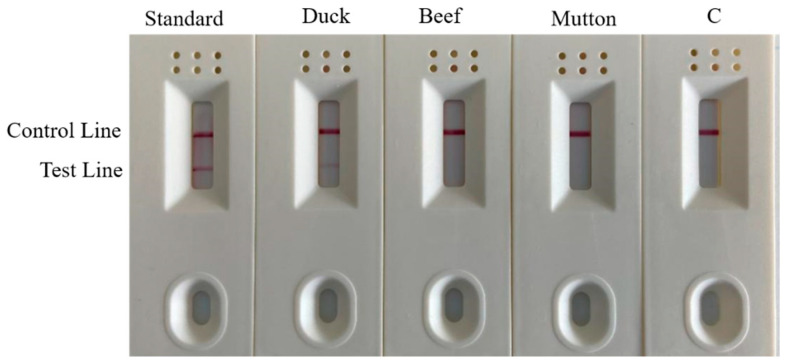
Strip test results. Lane C is the blank control.

**Figure 11 foods-11-01895-f011:**
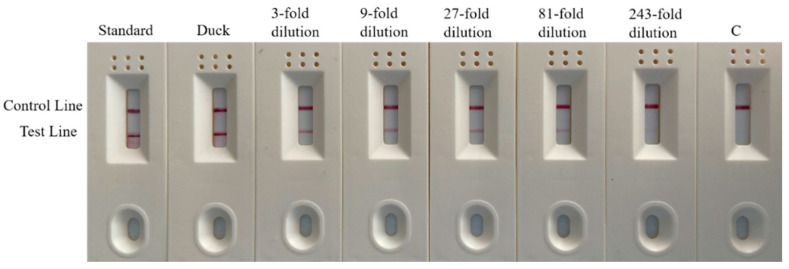
Test results of strip sensitivity. Channel C is blank control.

**Figure 12 foods-11-01895-f012:**
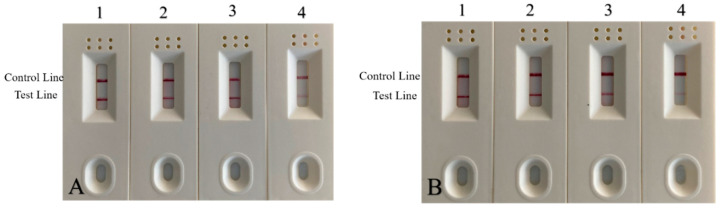
Strip test results of beef and mutton mixed with a duck by volume. (**A**) Results of beef and duck mixture. 1: 50% duck + 50% beef; 2: 10% duck + 90% beef; 3: 5% duck + 95% beef; 4: 1% duck meat + 99% beef. (**B**) Ratio results of mutton and duck mixture. 1: 50% duck + 50% mutton; 2: 10% duck + 90% mutton; 3: 5% duck + 95% mutton; 4: 1% duck meat + 99% mutton meat.

**Table 1 foods-11-01895-t001:** Composition of lysate.

Component	Final Concentration	Stock	Amount
NaCl	100 mM	1 M	1 mL
Tris-HCl	10 mM	50 mM	2 mL
EDTA, pH 8.0	25 mM	250 mM	1 mL
SDS	0.5%	10%	0.5 mL
Proteinase K	0.1 mg·mL^−1^	1 mg·mL^−1^	1 mL

Note: Sterile water was added to 10 mL; fresh proteinase K was added before use.

**Table 2 foods-11-01895-t002:** Sample preparation.

Duck Meat Percentage	50%	10%	5%	1%
Mixture of beef and duck	25 μL beef + 12.5 μL duck	25 μL beef + 2.5 μL duck	25 μL beef + 1.25 μL duck	25 μL beef + 0.25 μL duck
Mixture of mutton and duck	25 μL mutton + 12.5 μL duck	25 μL mutton + 2.5 μL duck	25 μL mutton + 1.25 μL duck	25 μL mutton + 0.25 μL duck

**Table 3 foods-11-01895-t003:** Sequences of Primers and probes.

Purpose	Name	Sequences (5′-3′)	Ref.
RPA	IL2-RPA-F1	CAACCAGAACACTGACAAGATGTGCAAAGTACTCA	This study
	IL2-RPA-F2	AGAACACTGACAAGATGTGCAAAGTACTCATCTTC	
	IL2-RPA-F3	ACTGACAAGATGTGCAAAGTACTCATCTTCAGCTG	
	IL2-RPA-F4	CAAGATGTGCAAAGTACTCATCTTCAGCTGCCTTT	
	IL2-RPA-F5	TGTGCAAAGTACTCATCTTCAGCTGCCTTTCAGTA	
	IL2-RPA-R1	ATCTTAGCAGTCCTGAAGACAGAAAACTTGAACTT	
	IL2-RPA-R2	AGCAGTCCTGAAGACAGAAAACTTGAACTTACCTT	
	IL2-RPA-R3	TCCTGAAGACAGAAAACTTGAACTTACCTTTGTGT	
	IL2-RPA-R4	AAGACAGAAAACTTGAACTTACCTTTGTGTCATTT	
	IL2-RPA-R5	AGAAAACTTGAACTTACCTTTGTGTCATTTGGTGT	
	IL2-RPA-P	TAGAAAACCTGGGAACAAGCATGXATGTAAGTGGATG	
PCR	F	GGAGCACCTCTATCAGAGAAAGACA	This study
	R	GTGTGTAGAGCTCAAGATCAATCCC	
	P	FAM-TGGGAACAAGCATGAATGTAAGTGGATGGT-BHQ1	

**Table 4 foods-11-01895-t004:** Primer-probe concentration.

Primer Final Concentration (μM)	Probe Final Concentration (μM)	Primer-Probe Final Concentration Ratio
0.4	0.12	10:3
0.3	0.12	10:4
0.2	0.12	5:3

**Table 5 foods-11-01895-t005:** Fluorescence RPA detection reaction system.

Reagent	Final Concentration	Volume
A buffer		29.4 μL
F	0.4 μM	2 μL
R	0.4 μM	2 μL
P	0.12 μM	0.6 μL
H_2_O		11.5 μL
DNA template		2 μL
B buffer		2.5 μL
Total volume		50 μL

Note: The reaction system can be adjusted appropriately according to the requirements of the instrument.

**Table 6 foods-11-01895-t006:** The comparison of dRPA and other detection methods.

Method	Reaction Time (h)	Instrument	Temperature (°C)	Operation	Point-of-Care
PCR	1.5	+	50–95	Normal	−
dd-PCR	2	+	50–95	Normal	−
LAMP	1	−	65	Normal	−
RPA	1	−	37–42	Normal	+
dRPA (This work)	0.5	−	37–42	Easy	+

Note: + means positive, − means negative.

## Data Availability

Data is contained within the article.

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
