# Peer review of "Development and Application of a Visual Duck Meat Detection Strategy for Molecular Diagnosis of Duck-Derived Components"

_foods, 2022, doi:10.3390/foods11131895_

Round 1

Reviewer 1 Report

Development and application of a visual duck meat detection strategy for molecular diagnosis of duck-derived components

This work addresses a simple, fast and inexpensive method for meat adulteration detection, which is a very relevant topic for food science with enormous practical potential. The experimental comparisons demonstrate effective detection of samples adulterated with 1% duck meat. The results are consistent, and the test strip is accurate and sensitive, which shows its potential for practical application.

The article is well organized and well written and, as such, is simple to follow. The text approaches the subject scientifically and is supported by accurate references. The Tables and Figures are essential for understanding the article, but some corrections are necessary to improve its clarity. For example, authors should consider standardizing the scale on the y-axis in Figure 5 and Figure 8. The material and methods are clearly described, which allows a perfect understanding of what has been done. The results are well presented; however, their discussion deserves further attention. Finally, the results corroborate the conclusions.

Author Response

1. This work addresses a simple, fast and inexpensive method for meat adulteration detection, which is a very relevant topic for food science with enormous practical potential. The experimental comparisons demonstrate effective detection of samples adulterated with 1% duck meat. The results are consistent, and the test strip is accurate and sensitive, which shows its potential for practical application.

Response:

Thanks to the reviewers for acknowledging our work. 

2.  The Tables and Figures are essential for understanding the article, but some corrections are necessary to improve its clarity. For example, authors should consider standardizing the scale on the y-axis in Figure 5 and Figure 8. 

Response:

Thanks for the suggestions. We agree that the scale on axis should standardized.  But the results showed in Figure 5 and 8 are special, so we think the format of y-axis are more reasonable.

Because, Figure 5 shows different reaction time of RPA. Within the short time, the result show good S/B (Figure 5B). Long time will trigger the second round of amplification (Figure 5a), so the fluorescent is higher. However, the figure do have some mistakes. Thanks for the suggestions again, we have modified the figure in MS.

In Figure 8, we do consider the standardizing the scale of Y-axis. We made the 8A and 8B the same scale, 8C and 8D the same scale.  Because, the 8A and 8B are results of RPA method. 8C and 8D are results of PCR method. In addition, all the fluorescent signal are read out by Bio-Rad CFX96 and show good S/B.

Reviewer 2 Report

The manuscript combine problem of adulteration of beef meat by duck meat with a duck-derived meat rapid detection. Critical is what authors undertook, threatening consumer health by such practices. The Authors established a rapid detection Recombinase Polymerase Amplification (dRPA) method by using interleukin 2 (IL-2) from nuclear genomic DNA as the target gene to design specific 16 primers.

Maybe, worthy of consideration is to prepare a Figure presenting the benefits listed in the abstract of the Recombinase Polymerase Amplification (dRPA) method.

For meat technologists also helpful would be presenting in the following Figure relation dRPA in comparison with other methods (DNA methods, including PCR, multiplex PCR, digital PCR, and RPA) to show the achievements of the Authors in finding the new approach, faster visual detection method based on RPA for detecting meat adulteration (some parts of Introduction). Moreover, what improvements were introduced (from conclusions).

Duck meat is poultry meat, and when we compare it with beef meat, it may be worth undertaking what kinds of muscles are in both meat of different species.

Please explain why you use dog meat- not applied in other countries.

Author Response

1. Maybe, worthy of consideration is to prepare a Figure presenting the benefits listed in the abstract of the Recombinase Polymerase Amplification (dRPA) method. For meat technologists also helpful would be presenting in the following Figure relation dRPA in comparison with other methods (DNA methods, including PCR, multiplex PCR, digital PCR, and RPA) to show the achievements of the Authors in finding the new approach, faster visual detection method based on RPA for detecting meat adulteration (some parts of Introduction). Moreover, what improvements were introduced (from conclusions).

Response: 

Thanks for the suggestions. We agree with the reviewer that table presenting the benefits of the dRPA method is needed. We have added a table in MS. For instance, 

Table 6. The comparison of dRPA and other detection methods.

Method

Reaction time (h)

Instrument

Temperature (℃)

Operation

Point-of-care

PCR

1.5

+

50-95

Normal

-

dd-PCR

2

+

50-95

Normal

-

LAMP

1

-

65

Normal

-

RPA

1

-

37-42

Normal

+

dRPA

(This work)

0.5

-

37-42

Easy

+

2. Duck meat is poultry meat, and when we compare it with beef meat, it may be worth undertaking what kinds of muscles are in both meat of different species.

Response: 

We agree with reviewers' opinions that the kinds of muscles of poultry meat and beef meat are worth to considering. But, our MS aim to design a rapid method for detection the meat issue, so we focus more on the difficulties of techniques not the muscle type. However, this is really a good questions, we will follow up this.

3.  Please explain why you use dog meat- not applied in other countries.

Response: 

Thanks for the suggestions. Regarding dog meat, we want to clarify that there is a custom of eating dog meat in some provinces of China, and the price of dog meat is high, which may be sold with adulterated duck meat by some Illegal merchants. So, we chose dog meat here to examine our method for duck meat detection. In addition, the dog meat material was purchased from the market.